# Particleboard Creation from Agricultural Waste Residue of Seed Hemp

**DOI:** 10.3390/ma16155316

**Published:** 2023-07-28

**Authors:** Kristaps Zvirgzds, Edgars Kirilovs, Silvija Kukle, Inga Zotova, Ilze Gudro, Uldis Gross

**Affiliations:** 1Faculty of Materials Science and Applied Chemistry, Institute of Design Technologies, Riga Technical University, LV-1658 Riga, Latvia; edgars.kirilovs@rtu.lv (E.K.); silvija.kukle@rtu.lv (S.K.); inga.zotova@rtu.lv (I.Z.); ilze.gudro@rtu.lv (I.G.); 2Department of Information Technologies, Latvia University of Life Sciences and Technologies, LV-3001 Jelgava, Latvia; fkgross@llu.lv

**Keywords:** hemp, hemp shives, hemp processing, agricultural residue, renewable resources, particleboard, particle size, board material

## Abstract

In this research, agricultural residue of seed hemp variety “Adzelvieši” was used to create hemp particleboard samples. Hemp was grown in three experimental fields where it was observed that after seed harvesting, 3.5 tonnes of hemp stems per hectare remained. The plants were processed with milling, cutting, and sieving equipment. Moisture content and particle size distribution were observed throughout raw material processing. Hemp boards were produced using the cold pressing method with 10% urea formaldehyde resin as the binder. The boards were made as 20 mm thick single-layer parts with a density range of 220 ± 30 kg/m^3^ and porosity of 86%. Board structural analysis was performed using optical microscopy and scanning electron microscopy methods. Mechanical strength was determined by performing bending strength, internal bond strength, and screw withdrawal tests. The thermal conductivity reached 0.047 ± 0.008 W/(mK). The results were compared with industrially produced hemp shive boards and materials in the developmental or production stage. The feasibility for the experimental production cycle proposed in the study is discussed.

## 1. Introduction

Over the past few decades, the utilisation of biomass resources has gained popularity as a method for producing board materials [1,2,3,4,5]. Biomass refers to organic matter derived from renewable sources such as wood chips, sawdust, various grain crop stalks, fibres, seeds, and other sources. The most valuable portion of these resources is the leftover waste generated after agricultural and forest management processes. Researchers report that the realistic potential of available agricultural crop residue, excluding grass, is 75 megatons each year in Europe. According to Eurostat and the European Court of Auditors, the European Union countries generated a total of 21.3 billion tonnes of agricultural waste in 2018, including waste from crops, fruits, vegetables, animal manure, and other agricultural activities [6,7,8]. Some of the generated waste can be used for energy production or recycled into useful materials [1,2,9,10]. To determine which agricultural waste sources can serve as primary ingredients for board material production, individual assessments must be conducted for each source. These assessments evaluate factors such as biomass sources, harvesting methods, processing options, environmental impact, cost management, and potential uses for the finished products, as demonstrated in numerous recent studies [4,5,11,12,13,14,15,16,17].

Hemp is considered one of the main sources of available biomass due to its fast growth, high yield, and low input requirements, making it a promising resource for various applications [18,19,20,21,22]. There are three main types of hemp crops grown by farmers—for fibre and shive production (industrial hemp), grain (seed) harvesting, and metabolite extraction. Hemp is grown across the world in various climatic conditions and regions—China, North America, South Africa, etc. In recent years, the area dedicated to hemp cultivation has increased significantly in Europe from 20,540 ha in 2015 to 33,020 ha in 2022 (a 60% increase) [23].

While industrial hemp growth and harvesting are conducted using equipment to utilise the whole plant, seed hemp farmers have limitations in hemp harvesting applications and equipment, and with a lack of suitable processing units, a significant portion of the plant—the stems—is left unused in the fields by farmers. At present, seed hemp farmers are facing an urgent problem related to plant stem disposal after seed harvesting, as the management of residue coming from the agricultural sector has become a global matter of concern. In Europe, agricultural waste processing for recovering bioactive compounds or developing innovative and green bio-based materials is in full agreement with the aim of the circular economy [24,25]. It is estimated that between 5 and 15 tonnes of hemp stems can be harvested per hectare of crops to obtain fibres and shives for manufacturing new products [26,27,28]. The exact available amount depends on growth conditions, variety, and other factors. For example, in Latvia, there are currently no hemp-specific processing factories despite at least two public attempts to build such facilities with projects in advanced stages [29,30]. A construction company that builds homes using hemp shive and lime biocomposite imports the raw material from France and Germany just because the locally grown hemp is not processed in a factory that is certified up to EU standards [31]. Hemp plant residue can be used in biofuel and biogas production but those applications need a reliable and constant raw material supply [32]. For low-scale farmers, there is demand for cheaper and locally available processing options. In 2022, by conducting interviews with hemp producers, it was understood that after the growing and harvesting phases of plants for the food industry, it is difficult to find use for leftover stems. Fibers twisting around harvesting equipment rotors, stem unsuitability for baling, and a lack of processing options lead to the fact that after the harvesting of seeds, the leftovers are ploughed into soil or burned instead of processing.

There has been research into alternative methods of hemp processing to bypass field retting or other traditional processes [33]. The produced hemp particles—shives and fibre—can be used to produce construction and board materials and enhance material properties by utilising them as main ingredients or additives [12,34,35,36]. Efforts are being made to address the environmental impact and health concerns related to the utilisation of hemp residues, aiming to maximise their positive effects. One active research direction in the production of board materials involves plant-based binders that replace formaldehyde compounds with lignin and lignin-containing compounds [37,38,39,40,41]. Other alternative binders include soy-based resins, tannin-based adhesives, and bio-based polymers [42,43]. By combining such binders with the properties of hemp, it becomes possible to develop materials with minimal or no negative impact on the environment throughout their life cycle. Even just with its natural features such as biodegradability, fast growth, and CO_2_ sequestration amounts, hemp as the main ingredient replacing wood chips gives additional value to products made using more traditional binders such as urea-formaldehyde (UF), melamine-urea-formaldehyde (MUF), phenol-formaldehyde (PF), and isocyanate-based resins. There are industrial hemp boards and blocks already available on the market made by producers such as IsoHemp, Hempearth, and Hespan [44,45,46]. 

Hemp shives have good thermal insulation properties. Materials that use them as the base material are measured to have the thermal conductivity values in the range from 0.054 W/(mK) to 0.094 W/(mK) [45,46,47,48]. Materials well known in the construction industry often possess lower thermal conductivity values—rock wool, glass wool, and phenolic foam have thermal conductivity values in the range from 0.018 W/(mK) to 0.040 W/(mK) [49]. These materials have disadvantages due to their energy-intensive manufacturing or lengthy decomposition of scrap and utilised parts. 

The value of hemp is that each part of the plant can make a positive contribution to the industrial sector if there is demand for the resource. The distance from the field to the processing site is very important in the modern world [50,51]. If grown locally at the small scale, it can be used to create interior panels or furniture, as shown by RikMakes in The Netherlands that creates biodegradable compost board products [52]. 

In terms of economic value, bast fibres are refined to different grades for products such as insulation, biocomposites, and textiles. For example, textile-grade fibres can be found in the 1200 Euro/tonne range, but hemp shives in Europe in 2022 range from roughly 200 to 450 Euro/tonne in various sources [53,54].

The objective of this study is to examine whether the residuals of seed hemp growers after seed harvesting can be processed by simple methods to later produce a board material that is close or equivalent in properties to board materials made from industrially prepared shives. If such a plant preparation process is determined to produce competitive material, it would provide an incentive for growers to collect, process, and offer this resource on the open market in regions where hemp processing facilities are outdated or non-existent. The objectives are to (a) determine how much available leftovers are produced after seed harvesting; (b) test raw material preparation and the board material production cycle from growth to creation of a product; and (c) produce board samples to identify density, structural integrity, bending strength, internal bond strength, screw withdrawal resistance, and thermal values.

## 2. Materials and Methods

The materials and methods in this study were approximated to the conditions of real medium-sized agricultural production (10 to 50 ha of arable land area), where hemp is one of several cultivated crops. The equipment used in the preparation of the material was chosen so that, with minimal investment, it would be available to producers of agricultural products who could use it to expand their product range. Processes that are further discussed in Section 2.1, Section 2.2 and Section 2.3 are stated in the scheme in Figure 1.

The methods used in the production and testing of board materials were selected in accordance with standards to determine the properties important for mechanical strength and application purposes. 

### 2.1. Material Preparation

The materials used in the experiment for analysing the situation from an agricultural standpoint and preparing the boards were obtained from hemp grown in 2022 in three places in Latvia—“Adzelvieši” in Burtnieki municipality, Latvia; “Stariņi” in Ogre municipality, Latvia; and Marupe municipality, Latvia.

Hemp was grown by sowing at the end of April (25–29 April) and harvesting at the end of September (15–30 September). The harvesting date is comparable to the situation of real growing and production conditions, where the hemp threshing time is affected by the degree of ripeness of the crop, the weather (rain, humidity), and the availability of equipment for the threshing process.

According to the Latvian Environment, Geology and Meteorology Centre, SLLC, Latvia data, the air temperature hourly average (HTDRY) was 16.2 °C, the monthly precipitation amount (HPRAB) was 60 mm, and the sunshine duration (HSND) was 293 h each month during the period of vegetation [55].

The hemp plants used for the board material samples were grown in “Stariņi”, Ogre municipality. The size of the experimental field was 0.3 hectares and the soil was categorised by type—sod-podzolic (ordinary) and soil acidity—5.1 pH. Cultivation took place using biological methods both in the preparation of arable land and in the cultivation of plants. Land preparation took place in the autumn of 2021 using a Kverneland 2 furrow plough (Kverneland group, Norway), incorporating locally produced organic fertiliser in the amount of 4.5 t/ha. In the spring of 2022, the soil was treated with a spring-tooth harrow ARAS 2.6 m (Rokiškio mašinu gamykla, JSC, Rokiškis, Lithuania) twice in a perpendicular manner. 

The “Adzelvieši” variety was used for the seed material, which is a certified seed hemp variety for plants used to obtain seeds. They were sown using a comb-type seed drill (see Figure 2A) with a seeding rate of 20 kg/ha. The seeds were embedded into soil using the same spring-tooth harrow. During the growth stage, the amount of weeds on the field was monitored, and it was observed that the second most commonly growing plant after hemp was wild radish (*Raphanus raphanistrum* L.). 

Harvesting of plants was conducted by hand using garden shears and a sickle. Plants were tied in bundles and put to dry in a storage facility so no additional moisture could be absorbed. Selected bundles were monitored by weighing them during the drying stage. After 3–4 months of storage, the plants were used for further processing described in Section 2.2.

To assess the available leftover amount, two sets of fully grown hemp plants of 1 m^2^ were gathered from locations in “Adzelvieši” in Burtnieki municipality, Latvia; and Marupe municipality, Latvia. Separately, the mass for stems and mass for panicles and seeds as well as stem length and diameter were measured according to the scheme in Figure 3. 

### 2.2. Material Processing

Two significant parts of hemp plants that are used to create or enhance various board and composite materials are fibres and shives. The hemp stem cross-section displays a clear distinction between the fibrous outer bast region and the woody inner shives region, with vascular bundles playing a significant role in resource transport and allocation within the stem. These parts are usually separated by decorticating in conjunction with retting (or other mechanical separation methods) to use either fibres or shives.

The process in which the plant is transformed into raw material suitable for board material production in this study was performed using methods that would be available to the farmer without making large investments in technical equipment, but by using what is already available in the agriculture establishment, such as branch shredders, grain mills, chipping equipment, or guillotines for various purposes. The materials produced from the whole hemp stem (Figure 4f–j) were compared with industrially prepared hemp shives for the board materials produced in the study (industrially prepared and sieved hemp shives shown in Figure 4a–e).

#### 2.2.1. Plant Comminution

In this study, two main equipment pieces were compared—a hammer-type mill and a branch shredder with a single 6-blade rotor to comminute the hemp stems (Figure 4f) into crumbled particles ready for board production. 

A MKU-T-3 hammer-type cutting mill (HIDROLAT SJSC, Liepaja, Latvia) with a 195 mm rotary chamber and two perpendicular hammers with 3 mallets on each side incorporating sieve inserts of 3.5 × 3.5 mm was used to produce milled hemp particles visible in Figure 4g, which, with sieving, were separated into two groups of milled fibres (MHF) and milled shives (MHS), as shown in Figure 4h,i, respectively. 

A MAKITA UD2500/2 branch shredder with a 6-blade rotor was used to comminute hemp stems. It runs on a standard 230 V electric motor with an output power of 2500 W and cutting speed of 40 rpm. The 120 mm cutting rotor presses the stems against a thick steel stopper by breaking the woody part of the stem into long pieces and cutting the fibre only partially, as shown in Figure 4j. This tool was chosen because of the affordable price ranging from 200 to 300 Euros in Latvia, 2023. It is a garden tool designed for shredding and chipping branches and garden waste, but, as observed in this study, it works just as well with hemp, and after several hours of shredding, the average productivity was valued to be about 30–50 kg/h depending on the number of blockages. Only the particle groups shown in Figure 4c,d,h–j were used to produce board samples stated in Table 1. These groups were the result of stem processing by either cutting or milling and sieving.

In comparison to the industrial preparation of hemp shives where hemp stems undergo retting and decortication to separate fibrous parts from the inner core and then use sieving or the air separation technique to sort particles, the given approach replaces these processes by obtaining a material where the shives and fibres are together in one mass. In Figure 4a, the material contains very small amounts of fibres, because these are industrially prepared clean hemp shives, which are separated into particle groups by sieving (Figure 4b–e) and contain almost no fibres at all as they are sieved off. In Figure 4f, the harvested hemp stems are shown, which were processed with the methods described in previous paragraphs—milling and cutting to create particle groups that contain a lot of fibrous substances (Figure 4g,j). The milled group was then separated with a sieve with 2 mm outlets to create groups MHF (larger than 2 mm, containing most of the fibres) and MHS (0.5–2 mm, containing mostly shives). The approach to use the whole hemp stem was influenced from research at the Leibniz Institute for Agricultural Engineering Potsdam-Bornim [56,57,58].

#### 2.2.2. Moisture Contents

To determine the raw material moisture contents, each sample group that was created was tested in accordance with the LVS EN 322:1993 standard, using the mass method [59]. Raw material from each type and each batch of milled or cut plants was placed in 3 separate aluminium foil containers. The containers were weighed with a Kern digital balance EMB 600-2 (to the nearest 0.01 g) and placed in a Binder EU280 heating oven. After 24 h at 105 ± 2 °C in accordance with the standard, each sample container was weighed to determine the change in mass. The moisture content of samples was calculated according to Equation (1) and the mean value of 3 samples was calculated. This test was conducted to monitor the quality of raw material and ensure moisture uniformity of the produced board samples.
(1)$$H=\frac{m_{H}-m_{0}}{m_{0}}\times 100$$
where H—moisture content of the raw material, percentage; $m_{H}$—mass of test specimen at normal climate, g; $a_{w}$, $m_{0}$—mass of specimen after drying, g. 

#### 2.2.3. Granulometric Analysis

Each raw material batch possesses a unique particle size distribution. As observed in a previous study, the small particles are harmful to the production of quality samples because the binder absorbs moisture too quickly. Therefore, particles smaller than 500 µm are excluded from the groups by sieving the material with a sieve opening of 500 µm. After undergoing the sieving process, the raw material is evaluated to visualise and compare the size distribution of the different preparation methods. The sieving test adheres to the LVS EN 933: 2012 standard [60]. By using this method, the smaller particles are separated from the larger ones, and it can be observed how much of the small particles are still present in material mass. 

To conduct the test, samples were collected from the storage location immediately after drying and placed in a controlled environment until testing. Prior to sieving, all raw material batches were divided into smaller portions, each weighing no more than 30 g. This partitioning was conducted to ensure more effective sieving and to reduce the occurrence of inaccurate results. The sieving process was carried out using a MATEST A059-01 electromagnetic sieve shaker (with sieve insert sizes of 5.6 mm, 3.15 mm, 2 mm, 1 mm, 500 µm, 355 µm, 160 µm, and 90 µm using an interrupted cycle with a vibration time of 58 s and an interruption of 2 s, and the total sieving cycle of each sample lasts 9 min at a vibration intensity of 80 W/m^2^). 

### 2.3. Board Material Production

Experimental mixtures were prepared according to a recipe consisting of 85% base material (hemp), 10% solid binder, and 5% water [47]. Table 1 shows the material groups that have been created and compared in this study are stated and named where HS-1 and HS-2 (HS—hemp shives) boards were created from industrial hemp shives from the *Biolabrezskie* variety (owner: Institute of Natural Fibres and Medicinal Plants, Poznan, Poland) as well as three new groups of Milled hemp fibres (MHF), Milled hemp shives (MHS), and Cut Hemp (CH), that were created from the locally grown *Adzelvieši* variety (owner: Institute of Agricultural Resources and Economics, Priekuli, Latvia and z/s Adzelvieši, Burtnieki, Latvia). The unique material preparation methods and equipment for each sample group are described in Section 2.2.1 and Section 2.3.1.

#### 2.3.1. Hemp Particles

The main ingredient in the board material was hemp stem particles. After harvesting and processing as described in Section 2.2, the material was sieved by hand using sieves from the MATEST A059-01 electromagnetic sieve shaker with the aim of removing particles smaller than 500 µm as they absorb most of the binder moisture during the mixing phase. Sieving to specific particle size groups allows for evaluating the importance of the particle size influence on the produced board properties. 

The moisture content of each particle group after sieving was monitored according to the LVS EN 322: 1993 standard. 

#### 2.3.2. Binder

For particle binding, a Urea Formaldehyde Resin (UF) Kleiberit 862.0 (Intarsija LLC., Riga, Latvia) was used as the adhesive. This product is widely used in the wood industry. It is mostly used for the adhesion of veneers on particleboard and its manufacture. This binder complies with the E1 emission class (emits less than 0.1 ppm of formaldehyde into the ambient air) [61]. It is possible to produce materials compliant with the standards EN 312 and EN 13986, which regulate specifications of particleboards and wood-based panels for structural use. It was chosen to create boards comparable with a previous study where industrially produced hemp shives were used as base material for the boards. The UF resin is usable in both hot and cold pressing conditions [47]. The technical data sheet of the binder states that the drying time at 20 °C is 4 h and 30 min; at 60 °C, it is only 6 min. The weight ratio for the mixture is 2:1 (powder: water) and the open time of the mixed binder is ~5 min.

One main disadvantage of the UF resin is hydrolytic degradation when in the presence of moisture, water, or acid. The produced boards will be water-permeable and can degrade or dissolve in a short period if used in humid conditions or environments.

#### 2.3.3. Board Production

Board samples were created using a hydraulic press with a mould of size 405 × 405 mm in cold pressing technique. A mixture was prepared using 912 g of hemp mass and 10 wt.% UF binder powder and 5 wt.% water, matching the volumes used in similar research [47,62,63]. 

The mixture of hemp particles and binder was created with an electric mixer in the case for groups HS-1, HS-2, MHF, and MHS. For particle group CH, the mixing process was conducted by simultaneously spraying the binder and mixing the mass by hand. This was conducted because the fibres that were not perfectly cut wraps around the mixer, creating a winding like a yarn does (average fibre length from 50 measurements was 47.2 cm). Afterwards, the mass was formed by hand into the mould and left for curing for 24 h at 21 ± 1 °C. The mixture was pressed using a pressure of 0.72 MPa. The board thickness was controlled by monitoring the pressure gauge and punch depth in the mould.

Immediately after extraction of the mould, the samples were weighed and kept under laboratory conditions for 14 days, observing the drying process and changes in weight using the Kern EMB 600-2 digital scale (accuracy 0.01 g) at the same time each day. Based on the moisture determination standard LVS EN 322 [59], the water moisture content in board materials was determined using the mass method. 

Boards of size 405 × 405 mm were cut into smaller pieces to determine their structural properties and to prepare them for mechanical and physical testing. Cutting was performed by using a FELDER K 700 S sliding table panel saw.

A cutting scheme was determined and applied so that the test pieces were accommodated for planned tests according to requirements in each respective standard [64,65,66]. (Figure 5a). A 300 × 300 mm piece was used for the thermal conductivity test, a 50 × 400 mm piece for the bending strength, and 50 × 50 mm pieces for the internal bond strength, screw withdrawal, and water absorption tests.

Test pieces were measured with a digital scale for weight with a precision to 0.01 g and with a digital calliper for size with a precision to 0.01 mm at each side of the board (Figure 5b). The density was calculated according to the LVS EN 323:2000 standard using Equation (2): (2)$$\rho_{w}=\frac{m_{w}}{a_{w}\times b_{w}\times l_{w}}=\frac{m_{w}}{V_{w}}$$
where $\rho_{w}$—board density at standard humidity, kg/m^3^; $m_{w}$—mass of the test specimen, g; $a_{w}$, $b_{w}$—length and width of the test specimen, respectively, mm; $l_{w}$—thickness of the test specimen, mm; $V_{w}$—volume, m^3^.

### 2.4. Structural Analysis

To objectively evaluate the mechanical and moisture absorption capabilities of the material, a complete understanding of the material structure is required, so an evaluation of the bonding quality of the particles and the binder dispersion at several levels was conducted.

#### 2.4.1. Optical Microscopy

A Motica SZM171 TL stereomicroscope was used to analyse the structural uniformity and particle placement within boards. A total of 5 samples of each produced board material group were observed under 15× optical magnification on their surface, in crosscut view, and on the surface/side edge where the cut was made. 

#### 2.4.2. SEM Microscopy

The SEM (scanning electron microscopy) microscopy was performed at the Rudolfs Cimdins Riga Biomaterials Innovations and Development Centre of Riga Technical University (RTU, Riga, Latvia).

Board samples were prepared by cutting about 4 × 4 × 1 mm pieces from the surface, the inner part, and the cross-section of the boards. They were glued to sample holders using adhesive tape. Before scanning, the samples were prepared using a LEICA EM ACE200 vacuum coater (LEICA, Wetzlar, Germany) by applying a carbon coating using the multi-thread option with 15 pulses. The scanning electron microscopy analysis of samples was performed with a TESCAN Electron Microscope VEGA 4 (TESCAN, Vega 4, Brno, Czechia). Data analysis was conducted on TESCAN’s Essence™ software. 

#### 2.4.3. Pycnometry

The density of the material was determined by a helium pycnometer (Quantachrome, Ultrapyc 1200e, Boynton Beach, FL, USA). The large cell with a volume of 135 cm^3^ was used with the “repeat run” measurement mode until the results fell within a user-specified tolerance of ±0.1%. 

The board samples were milled in a coffee grinder and sieved manually with 355 µm outlets to prepare a powder-type substance suitable for a helium pycnometer. For each test run, about 0.6 g of specimen was placed within the cell and the mass was weighed with an accuracy of 0.001 g. The cell was then placed in the pycnometer and after a successful run, the reported data were analysed in Microsoft Excel for Microsoft 365 MSO (Version 2306) software.

The porosity of samples was calculated by comparing the bulk density of the material and the true density measured in the pycnometer. 

### 2.5. Mechanical Strengths

To determine the mechanical resistance of the board material for use in furniture production or the construction industry, the 3-point bending strength, the internal bond strength, and the pull-out force of screws were examined. The results were classified according to the LVS EN 312 standard [67].

All statistical calculations, the experimental design, and the processing of test results were performed with the Data Analysis ToolPak add-in on Microsoft Office 365 (Version 2306).

#### 2.5.1. Bending Strength

The bending strength evaluation was conducted with a FORMTEST UBP 86/200 universal testing machine according to the LVS EN 310 standard [64]. The load properties were measured on 50 × 405 mm sized board samples. The length of the samples was reduced to accommodate the production and testing capabilities of the machine. To calculate bending strength values from load measurements, Equation (3) was used:(3)$$\sigma_{b}=\frac{3F_{max}l_{1}}{2bd^{2}}$$
where $\sigma_{b}$—bending strength, MPa; $F_{max}$—maximum load, N; $l_{1}$—distance between the centres of the supports, mm; b—width of the test piece, mm; d—thickness of the test piece, mm.

For this test, at least five samples from each board material group were prepared and tested for the maximum load read and collapse point observation. While testing the samples, the deflection in mm with an accuracy of 0.01 mm was read with a GIMEX GIMS480 digital indicator. The maximum load value was used for comparison with other materials, and the read was taken from the testing machine. The collapse point was measured from the end points of the test piece to determine the possible offset, indicating that the material was unevenly distributed in the pressing stage. 

#### 2.5.2. Internal Bond Strength

The internal bond strength evaluation was conducted using a FORMTEST UBP 86/200 universal testing machine according to the LVS EN 319 standard [68]. 

The board pieces were cut to 50 × 50 mm and glued between two sides of aluminium sample holders with a hot glue gun. The glued samples were placed in the testing apparatus and, by gradually applying pressure, the maximum load was measured in kN. The mean value was calculated from 5 samples within each sample group. 

Samples were loaded until total separation into two pieces and pictures were taken for each sample to later analyse if the breakup occurred in the middle of the sample or was offset. By placing the pictures in AutoCAD 2022 (S.51.0.0) software and drawing vector lines above the pictures, a scheme was drawn to compare the breakup line offset between groups of samples. 

#### 2.5.3. Screw Withdrawal

The screw withdrawal resistance was tested using a FORMTEST UBP 86/200 universal testing machine according to the LVS EN 320 standard [69].

From each board group, 5 samples were prepared measuring 50 × 50 mm. By drawing two perpendicular lines from test piece corners, the centre was marked where a 4.5 × 40 mm screw was inserted into the sample. It was decided not to perform a predrill, as the material density did not cause any surface extrusion around the screw or beneath it. The test pieces were placed within the testing machine and, by gradually applying force, the maximum load in kN was measured for each sample. Mean values were calculated for each sample group. 

### 2.6. Thermal Conductivity

To compare the developed material with similar agricultural residue particle and wood particle board materials, the thermal conductivity coefficient was measured with a NETZSCH HFM 446 Lambda heat flow meter according to the LVS ISO 8302:2001 standard [66] by testing 300 × 300 mm board pieces at a user-defined temperature difference between 0 °C and 20 °C. Thermal conductivity was calculated according to Equation (4):(4)$$\lambda =\frac{Q}{A}\times \frac{L}{∆T}$$
where *λ*—thermal conductivity, W/(mK); *Q*—amount of heat transferred through the material, W; *L*—distance between the two isothermal planes, m; *A*—area of the surface, m^2^; ∆*T*—temperature difference, K.

The results were compared to other materials with similar density values or similar binding components.

### 2.7. Production Feasibility 

To determine whether the idea is economically and practically viable to give an actual contribution to agricultural producers, the price of hemp resources was first examined, both for fibres and shives as well as the finished products, with which the boards would have to compete. The price review mostly considered the data available on the European market about official dealers because the data are collected on a variety of hemp selected and grown in Europe. Secondly, board materials and products already available on the market or design prototypes made from agricultural residues were examined.

For experimental purposes, moulds were developed to produce two different decorative containers by pressing hemp shives according to the same recipe from which the board material samples were made. The moulds were made of birch plywood. To prevent the hemp mass from sticking to the plywood, it was wrapped in a thin polyethylene film for the duration of the pressing.

## 3. Results

When using the whole hemp stem, some of the mechanical production processes of decortication such as field-retting, mechanical breaking, separating, and sorting can be substituted with much simpler cutting or milling methods to produce the raw material for board manufacturing. On the condition that the properties of the boards created using all parts of the hemp stem—the fibres and shives without separation—reach equivalent or close properties to boards made from just shives, it would encourage agricultural producers to redirect this biomass resource for the creation of new products. Using the whole hemp plant for board material creation is not unique approach, with prior research looking into adjustments of hemp preparation methods to create boards for furniture production [70,71]. This research looks into the simplification of hemp plant processing for areas without designated processing units and farmers without state-of-the-art combine harvesters adjusted for hemp plant harvesting. 

### 3.1. Material Properties

An assessment was conducted on three hemp crop fields, specifically the “Adzelvieši” hemp variety, and it was determined that approximately 3.5 dry tonnes per hectare of hemp stems can be collected after the hemp seed harvesting process in the climatic conditions of Latvia, as shown in Table 2.

It was observed that as the plant density per square meter unit increases, the plant height and stem diameter decreases. As stated in the literature, the hemp planting density has little effect on seed harvest amount, but with much higher planting density, it is easier to harvest hemp for both seed and biomass [72]. Plant density has the largest influence on what the raw material for board production will consist of—as there are more plants per square metre, they are thinner and the prepared shives should have a smaller diameter.

It was estimated that for a field of hemp with an average plant length of 138 cm, the total mass is about 18 t/ha from which at least 6 t/ha of stems are left unused. 

Drying shrinkage and weight loss may vary due to weather conditions in the days prior to the harvest. The distance between the experimental field sites is around 100 km, so a small variation in rain and temperature can influence that. Another aspect is that the thinner stems have less moisture to evaporate and therefore shrink.

To store the harvested stems, it was decided not to leave them on the field as traditional methods suggest for field-retting purposes, because the seed producers need to perform soil preparation in the same autumn when harvests take place. This allows the material to be stored in hay roll or grain straw bale storage facilities. 

As explained in Section 3.2, it also benefits the CH sample preparation method where the fibre is not fully separated from the shives and less material crumbling occurs after the production of board samples. Both - hemp milling (MHF, MHS samples) and cutting (CH samples) present good work efficiency in material preparation as the milling equipment allows for processing about 25–40 kg/h, and the cutting equipment allows for processing 30–50 kg/h. Both might improve if higher-capacity and more costly machines are used. Higher-power equipment might reduce the number of blockages for rotors that are caused by hemp wrapping around them or dust particles clogging up the sieve of the mill. This would increase the efficiency of processes even more. Milling produces a lot more dust particles than cutting—18% in comparison to 2%, as shown in Figure 6. The granulometric analysis cannot show fully accurate results for coarser particles of CH, because some of the stems are not fully cut into small pieces, just broken into shorter segments that are still held together by the fibres. 

During the study, at least five boards from each type were produced to ensure equality within the sample group. Mean thickness and density values of board types are displayed in Table 3 to track the types of boards and their properties after reaching moisture equilibrium. 

For each conducted test, a mean density value was calculated and shown according to the specific test piece of the board used. The density varied from side to side of the produced board samples due to the particle distribution by hand in the mould. More free-falling shives in HS-1 and HS-2 samples were more uniformly distributed than the fibre-consisting types—MHF, MHS and CH, where the fluffy fibre particles tended to interlock even before pressing and create larger fluffs. 

When pressing the boards, the target thickness of 20 mm and target pressure of 0.72 MPa were monitored. With coarser particle groups (CH, MHF), it was chosen not to exceed the target pressure, to remain within the press limits; therefore, a difference in thickness was observed. To achieve boards of uniform density under production conditions, it may be necessary to adjust the pressing force depending on the batch of prepared material if particle screening has not been carried out.

Immediately after extraction from the mould, the calculated moisture contents of board samples were 20.7 ± 1.8% of the total board mass. The board thickness would increase minimally in the moments after extraction due to the pressure relief, but would then decrease in thickness and size along with this loss in moisture content. By monitoring board sample moisture, it was observed that during the first 5 days of drying at 21 ± 1 °C, the samples lost most of their moisture, and as the days passed, the more brittle samples started dropping particles on the surface after picking them up for measurements. It was also observed that after 10 days of drying, the moisture content stabilised and the samples were ready for further evaluation and testing.

### 3.2. Structural Analysis

At first glance, samples using only hemp shives were visually lighter and more uniform in surface (Figure 7b), while samples from the whole hemp stem were darker, with chaotically arranged fibre lines, as shown in Figure 7a. The surface for boards with both shives and fibres was rougher than for boards with just shives. The shive and fibre particles dried at different speeds and their volume shrinkage was different. In places with fibres in larger quantity, the board surface was bumpy.

The sample structure was initially observed by making a visual and touch assessment, and then more thoroughly under 15× optical magnification, which is shown in Table 4. Optical microscopy and scanning electron microscopy are only possible with a limited field of vision due to the change in sample height and particle dimensions. A significant correlation can be seen with material structural integrity just by cutting test samples. A lot of dropouts and crumbled edges were created during the cutting process for the smaller fraction (Table 4, b and c, HS-1 and MHS). The coarser particles (Table 4, b and c, HS-2, MHF, CH) were better bonded and interlocked with each other and so were cut rather than pulled out by the sawblade.

After careful observation test samples MHS and MHF were evaluated to be too brittle for practical use as they crumbled after several relocations from drying to weight measurement on the scale and back, so they were excluded from further testing of mechanical strength. The brittleness was most likely caused by low density and the use of UF resin, which are known to impart brittle bondlines, especially at higher bondline thicknesses, which was the case here due to the low density. To counter that, a more flexible adhesive or gap filling adhesive, such as PMDI-based, would be beneficial for specifically these particle groups. Another aspect for the low integrity of MHS and MHF boards could be the small fibres and shives size that have large surface area and need high amounts of adhesive for bonding. The percentage of binder used in fibreboard production, where particles have larger surface area, is generally higher (10–25%) than in particleboard (6–12%). 

Distinctive microstructures were observed through scanning electron microscopy. Despite the variation in particle size and preparation methods, the samples displayed a minimal variation in pore size and lacked a discernible arrangement of pores. 

A significant difference was observed for the surface quality of samples. After milling, the shives had significant abrasion markings, as visible in Figure 8C. In contrast, after cutting the material, the surface was naturally rougher with significant fibre splitting near the cut (Figure 8D). This might have been caused because the cutting blade was blunt and broke the stem apart by squeezing and tearing. 

The true density of hemp shive raw material that was later used in the production of HS-1 and HS-2 boards was measured as 1400 kg/m^3^; that for the hemp fibre that was used in the production of MHF was 1385 kg/m^3^ with a porosity of about 95%. The measured porosity for HS-1 and HS-2 that consisted of shives was 80.6%, whereas that for denser MHF boards was 72.8% (Table 5). The literature reports values of porosity around 76 to 80% for untreated hemp shive materials. The density is also close to the reported data on the cell-wall density and porosity of wood determined by gas pycnometry, so, in theory, it should be possible to substitute wood chips with hemp shives and maintain the durability and strength properties of board materials [73].

### 3.3. Mechanical Strength

Mechanical strength evaluation was conducted on groups HS-1, HS-2, and CH after the structural observation and exclusion of MHS and MHF from further evaluation (Table 6). A visual comparison for sample groups is presented in Figure 9. 

During the three-point bending strength (MOR) evaluation, it was observed that for HS-2, sample decay occurred at the lower surface (tension zone) near the centre mark, in some cases being offset from 0 to 5 mm off-centre. In 40% of cases for HS-1 samples, the decay occurred from 0 to 5 mm off-centre, but in the rest, 5 to 15 mm away from centre mark, meaning that there were spots of lower binder spread. Samples from the CH group broke apart in various parts of the 405 mm long sample, looking more like layers of hemp stems breaking apart (shear break) than a clear perpendicular break line as in the case of HS-1 and HS-2. After careful observation, it was clear that the CH samples were breaking apart in areas where a larger quantity of fibre particles were present within the board. 

The CH board resistance to three-point bending was calculated to be 3.5 ± 0.55 MPa. For the larger particle size group of HS-2 boards, it was 2.5 ± 0.1 MPa, and for the smaller particle size group of HS-1 boards, it was 2.4 ± 0.34 MPa (Figure 10). Despite an uneven structure and offset decay, the CH boards incorporating fibres were 33% better than that of HS-2, with HS-1 being an additional 3% worse. Such a relationship suggests that particle size and binding quality have a much lesser influence on bending strength than the addition of fibre in boards. It must be noted that the rupture of samples was occurring in parts where the fibre was in a larger quantity than the mix of fibre and shives or just shives, so the better resistance value cannot be due to the fibres themselves as a stronger component but fibres that improve the whole structure of the board. In comparison with similarly produced materials, the board bending strength values are higher than those of materials with similar density but a different binder used [19,74], but lower or similar to materials with a higher density range [75,76].

When calculating the modulus of elasticity, the HS-2 boards averaged 338.7 ± 58.2 MPa, HS-1—197.2 ± 31.7 MPa, and CH—231.4 ± 90.6 MPa.

The screw withdrawal test showed that boards cannot withstand large forces pulling the screws out. HS-1 reported a screw withdrawal strength of 10.6 ± 1.6 N/mm, HS-2—the best of 21.9 ± 4.1 N/mm, and CH—13.5 ± 2.6 N/mm. According to LVS EN 622-4, the resistance to screw axial withdrawal from wood-based fibreboards should be at least 30 N/mm. Even the best HS-2 sample group was 31.2% lower than specified in the standard. In comparison to material properties reported in other studies, the produced boards are in similar margins (from 12 to 20 N/mm) [76]. The reported values suggest that such boards can only be used as non-load-bearing material or need a layer of lamination to increase resistance to withdrawal. 

The internal bond testing showed different board gradations to the bending strength test. In this examination, HS-2 fared as the best board type, withstanding 0.12 ± 0.02 MPa of stress, with the HS-1 group coping with almost 40% less stress—0.08 ± 0.017 MPa, and CH—0.08 ± 0.05 MPa. These results coincide with reports where coarser-particle-size mixes have better internal bonding [77]. The produced boards have lower values than 0.2 MPa that is the requirement for 20 to 25 mm thick particleboard according to LVS EN 312:2011 specifications. The internal bonding requirements in EN 312 are based on the board thickness but also only valid for boards of “standard” density, which can be assumed to be in the range from 400 to 800 kg/m^3^. As the produced boards are in the range from 200 to 230 kg/m^3^, there is no definite reference with which they should comply. The reported values for particleboard with a capacity of utilisation are in the range from 0.3 to 0.5 MPa [78,79].

Even more than the measured forces, the collapse line that was close to the sample holders for each sample indicated a non-uniform structure and binder dispersion (Figure 11). For the HS-2 group, the break line was about 60 ± 5% offset from centre of the sample. Considering the method of particle forming in the mould, it is difficult to ensure an even fraction distribution, and it may occur that in one of the layers of the total thickness, there are particles with a smaller surface area or less binder application and so lesser adhesion. The test implementation is based on and adapted to the LVS EN 312 test method [67].

### 3.4. Thermal Conductivity

The thermal conductivity coefficient was measured for CH group samples and compared to HS-1 and HS-2 boards measured previously. For the three CH board samples with broad density properties, the average calculated λ is 0.048 ± 0.001 W/(mK) with a relative error of 3.8% (Figure 12). The average value calculated from the 8 HS-1 and HS-2 λ measurements (Figure 10), estimated with a relative error of 3.7%, is λ = 0.057 ± 0.002 W/(mK). The CH sample group is 17.5% better than those of the HS-1 and HS-2 groups. 

In practice, the thermal conductivity λ does not depend on the variance in density and thickness, but more on the variance in particle size and base material type, as boards with fibre and shive composition, and different preparation methods show results that are better by 17.5%. The differences in λ-values within each group shown in Figure 12 are statistically insignificant.

By comparing the produced materials to other agricultural and wood chip materials with similar density, the experimental samples have a lower or similar thermal conductivity [45,47,48,80,81,82,83]. The produced CH boards are very close to raw HS thermal conductivity values. The differences seen in Figure 12 originate mainly from the particle size range and binder distribution and clear air pockets that are important to slow down the heat flow. In the compared materials, the type of binders used is formaldehyde resin glue. 

### 3.5. Production Feasibility

Utilising agricultural waste as a raw material for particleboard helps address the challenges of waste management and reduces the need for landfill disposal or open burning, which can contribute to environmental pollution. By diverting agricultural waste from traditional waste streams, the environmental impact is minimised and the waste is transformed into a valuable resource. Waste resource management may sound easy and clear-cut in theory, but in practice, where additional costs are generated, time is consumed, additional equipment is needed, and the reality is more complex. A clear roadmap of how to manage these resources, how to harvest them, and how to add value must be presented to expect the agricultural production personnel investment in such a course. By giving guidelines, they can start investing in and utilising new equipment each year their production processes are planned. Additionally, a reliable processed material utilisation stream—must be made available. If board production streams are not available yet, a scenario where the processed raw material is used for animal or plant bedding is also viable. It is still a better use of resources than just disposal. 

According to the LLKC’s (Latvian Rural Consultation and Education Centre) 2022 gross coverage calculations, the average price for hemp stems produced for fibre production in Latvia is 150 EUR/t [84], so if seed producers were to harvest the leftover stems, they could increase their income by 525 Euro for each hectare harvested just by harvesting the stems. If the approach to process the hemp stems with cutting equipment is chosen, the price for the end material could go up to 3 times as the current price for clean hemp shives is about 400 EUR/t and for separated fibre is 912 EUR/t. The total for machine and manual labour operation for 1 ha of hemp cultivation (soil milling, ploughing, cultivation, sowing (hand labour), embedding seeds in the ground, harvesting seeds), taking into account the cost of fuel and seed material (6 EUR/kg), is 389 EUR/ha. These costs could be recouped by the income generated from stem processing. 

This study presented a low-scale approach of hemp leftover processing. To increase volumes and possibly improve the quality of the raw material produced, larger investments can be made, such as the purchase of specialized hemp decortication equipment such as the HurdMaster MD 1000 micro-decorticator for medium-scale production [85]. It could improve the output estimate from 30–50 kg/h to 50–100 kg/h. The approach presented in the study could be beneficial to more hemp producers that are growing other seed hemp varieties such as “Finola”, “Henola”, and others in regions where hemp fibre and shive processing factories are not built yet. 

During the project, two plywood moulds—203 × 203 mm—were made to produce prototypes for containers and/or panels with a curved relief made by cold pressing technique using the coarser HS-2 as filler. The density of the manufactured products was about 275 kg/m^3^ and the height was 16 and 34 mm (Figure 13). It was confirmed that this material recipe with minor tweaks is easily transferable to design projects to make dishes, wall and ceiling panels in unique shapes. To make such products of high quality and to be able to use them practically, the particles crumbling off edges must be reduced. This could be achieved either by increasing the amount of binder in the material recipe or by applying a protective coating such as a water-based varnish.

## 4. Conclusions

In this study, the use of a sustainable agricultural production leftover—seed hemp stem biomass—in board material production was researched to assess the feasibility of leftover utilization and create a roadmap to reduce its disposal. The board material avenue was chosen as a promising approach that could help meet the growing demand for eco-friendly and socially responsible products.

For this experiment, the local seed hemp variety “Adzelvieši” was used. It was grown in three experimental fields, cultivated, and harvested by biological means. The yield of hemp plants was about 17 t/ha, where 40% was leftover stems. With 49% of mass loss during drying, the available dry mass for processing was in range from 3.43 to 4.35 t/ha. Two hemp stem pre-processing technologies were used—milling and cutting. Milling produced a lot more dust particles than cutting—18% in comparison to 2%. In the case of cutting—some of the stems were not fully cut into particles, just broken into pieces that were still held together by the fibres. These processes avoid the field-retting, decortication, and sorting processes of industrial hemp to obtain clean shives but instead uses the whole stem. 

Boards from five prepared particle types (HS-1, HS-2, MHF, CH, MHS) were produced in cold pressing technique with 10% UF binder and a mean density ranging from 230 to 285 kg/m^3^. The moisture content stabilised within 10 days of drying the boards.

The structural analysis of the material was conducted using visual and touch assessment, with optical microscopy at 15× optical magnification, scanning electron microscopy, and a pycnometry. It was observed that the materials with more fibre in board contents were rougher in surface. After milling, the shives had significant abrasion markings, while after cutting the material, the surface was naturally rougher with significant fibre splitting near the cut. The calculated porosity of boards was in the range from 80.6% to 86.1%. Structural evaluation led to the exclusion of samples MHS and MHF because they were too brittle and crumbling for practical use.

The mechanical strength tests showed that CH boards were better than other fractions with the calculated bending strength resistance of 3.5 ± 0.5 MPa, which was due to the long fibre–shive particle structure. The HS-2 boards withstood 0.12 ± 0.02 MPa of tensile strain with a break line about 60 ± 5% offset from the centre of the samples, which indicates a non-uniform structure of materials. The testing of screw withdrawal from material reported 10.6 ± 1.6 N/mm of strength for HS-1 boards while HS-2 produced the best result at 21.9 ± 4.1 N/mm. The reported values suggest that such boards can only be used as non-load-bearing material as the values do not reach the minimum requirements for EN 312 load-bearing panels. 

The thermal conductivity value for the produced board material was 0.048 ± 0.0008 W/(mK) for shive–fibre boards (CH) in comparison to just shive boards—0.057 ± 0.002 W/(mK). Considering that the thermal conductivity of the created boards is comparable to the material in bulk form, it can be assessed as a positive aspect that the material can be attached to the wall and is stable in its form. With the aim of making environmentally friendly insulation material, the greatest gains are CO_2_ sequestrating raw material and low-energy consumption in production processes in comparison to traditional insulation materials such as mineral wool. 

According to the results, the created board material is suitable as an insulation material but, by reinforcing it within the framed panel structure, could be used as a space-separating element. A combination of methods—layer of lamination, a different choice of binder that fills the gaps between particles, and increased force when pressing samples to achieve higher density (400–600 kg/m^3^)—could increase the mechanical strength results and change application possibilities so that boards could replace low-density wood particleboards or could be used as material for light-weight furniture production. In a future study, increasing the board density to at least 400 kg/m^3^ (the lowest limit for particleboards to be classified as general-purpose boards for interior use with low-density according to EN 312) must be performed to confirm such proposals.

It was confirmed in this study that the proposed material preparation method and mixture recipe with minor tweaks is transferable to design projects to make dishes, wall and ceiling panels in unique shapes and forms are made. The produced board material could be used for a non-load-bearing applications such as a part of an interior-room-dividing wall.

## Figures and Tables

**Figure 1 materials-16-05316-f001:**
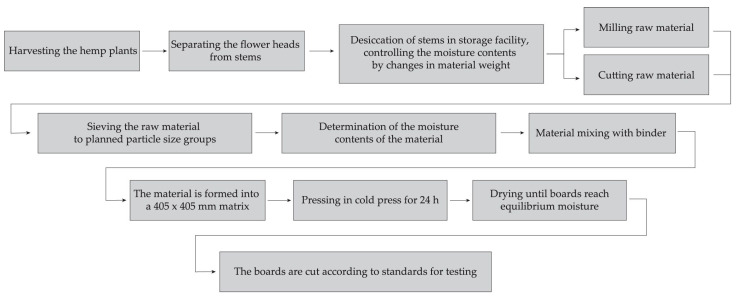
Scheme of growing, harvesting, processing, and production processes carried out during the study.

**Figure 2 materials-16-05316-f002:**
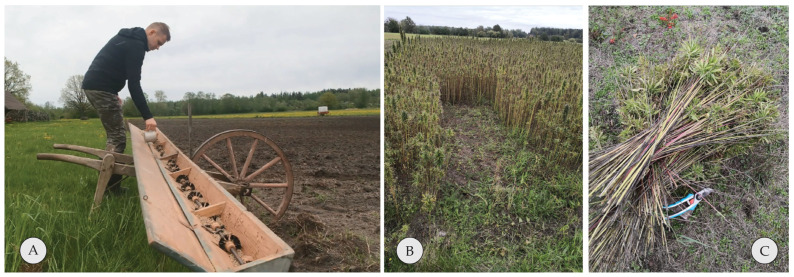
(**A**) Seeding stage, equipment used; (**B**) field during harvesting in September, (**C**) harvesting by hand using garden shears and sickle to cut the stems near ground.

**Figure 3 materials-16-05316-f003:**
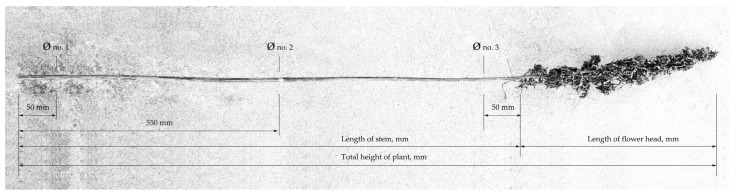
Hemp plant measurement scheme.

**Figure 4 materials-16-05316-f004:**
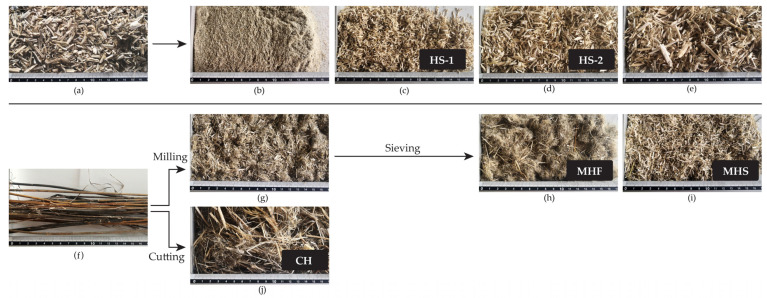
(**a**) Industrial hemp shives (HS), variety “Biolabrezskie”; (**b**) HS particle size less than 0.5 mm (industrially prepared (IP)); (**c**) HS-1 particle size range (0.5–2 mm) (IP); (**d**) HS-2 particle size range (2–5.6 mm) (IP); (**e**) HS particle sizes over 5.6 mm (IP); (**f**) harvested hemp stems, variety–*Adzelvieši* (CH); (**g**) milled hemp stems; (**h**) milled hemp stems—fibre particles (MHF); (**i**) milled hemp stems—shive particles (MHS); (**j**) cut hemp stems.

**Figure 5 materials-16-05316-f005:**
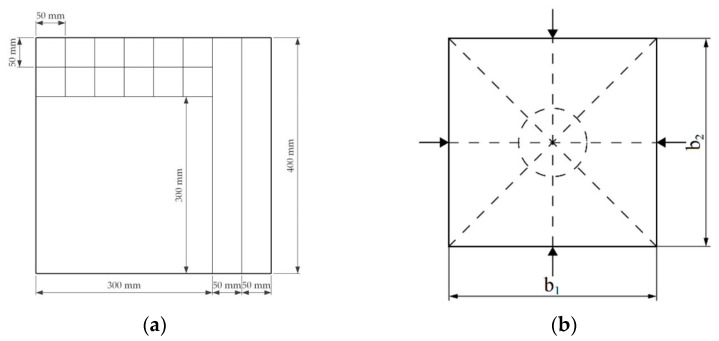
(**a**) Cutting scheme for produced boards. (**b**) Measurement scheme for determining the thickness of the samples [12].

**Figure 6 materials-16-05316-f006:**
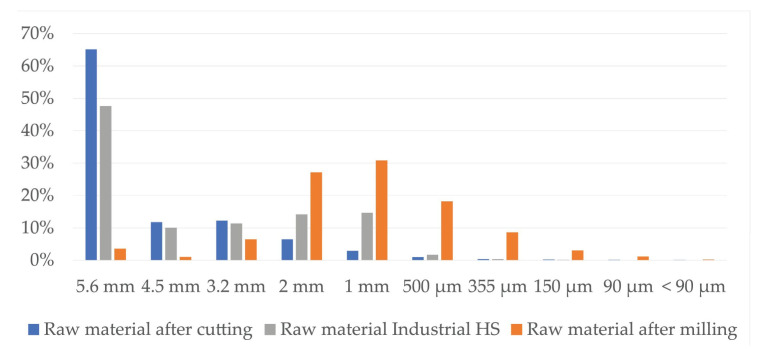
Granulometric analysis for raw material after processing.

**Figure 7 materials-16-05316-f007:**
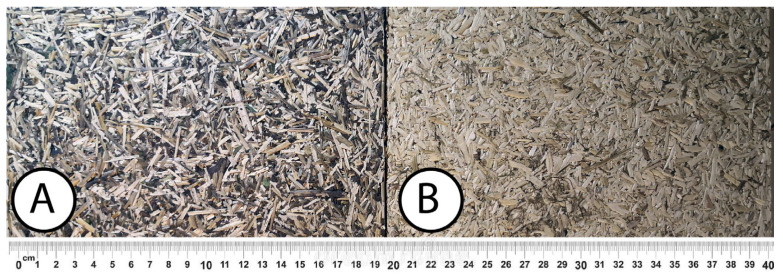
(**A**) Whole hemp stem particle board sample, (**B**) hemp shive board sample.

**Figure 8 materials-16-05316-f008:**
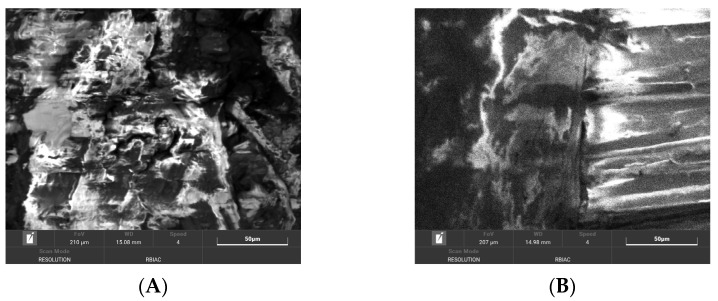
SEM of produced board and raw material samples; (**A**)—HS-1 sample, (**B**)—HS-2 sample, (**C**)—MHS sample, (**D**)—CH sample.

**Figure 9 materials-16-05316-f009:**
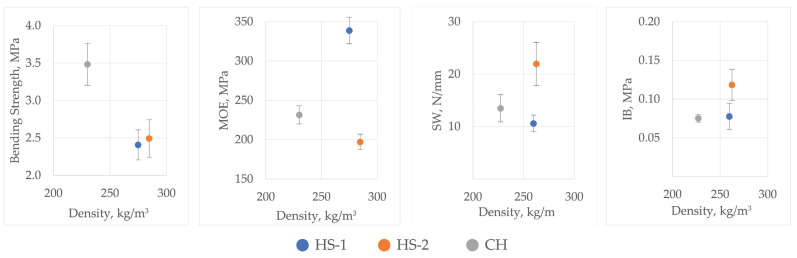
Bending strength, modulus of elasticity (MOE), screw withdrawal, and internal bonding values of HS-1, HS-2m and CH samples.

**Figure 10 materials-16-05316-f010:**
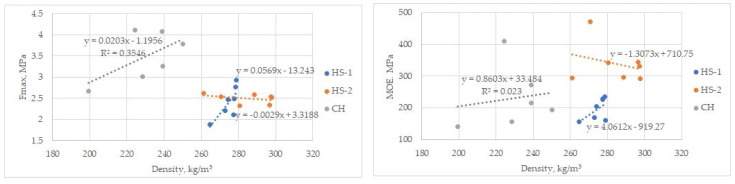
MOR and MOE for boards HS-1, HS-2, and CH.

**Figure 11 materials-16-05316-f011:**
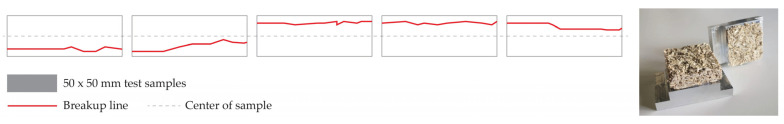
Internal bond testing results—median break line positioning within samples.

**Figure 12 materials-16-05316-f012:**
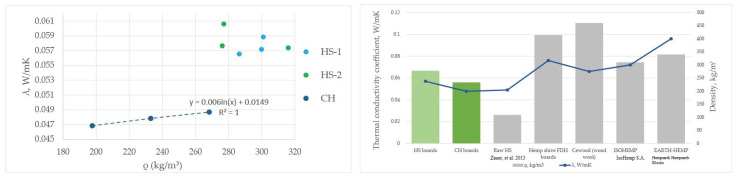
Thermal conductivity values for hemp shive test samples and comparison to similar materials [45,46,73].

**Figure 13 materials-16-05316-f013:**
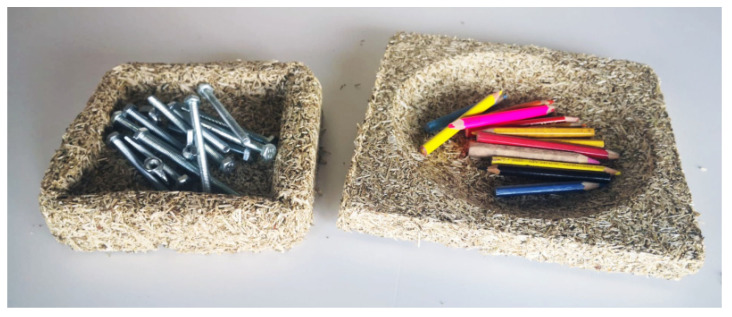
Hemp shive material dishes produced in specially created plywood moulds.

**Table 1 materials-16-05316-t001:** Classification of the hemp shive boards produced in the study.

Type	Particle Size (mm)	Tests Conducted *					
		SA	BS	IB	SW	TC	P
HS-1 **	0.5–2	X	X	X	X	X	X
HS-2 **	2–5.6	X	X	X	X	X	X
MHF	0.5–5.6	X					
MHS	0.5–5.6	X					X
CH	0.5–18	X	X	X	X	X	

* Explanation for abbreviation: SA—structural analysis, BS—bending strength, IB—internal bond test, SW—screw withdrawal test, TC—thermal conductivity, P–pycnometry. ** Bending strength, thermal conductivity values of HS-1, HS-2 have previously been reported in article “Production of Particleboard Using Various Particle Size Hemp Shives as Filler” [12].

**Table 2 materials-16-05316-t002:** Harvested plant yield data.

Experimental Field	Number of Plants on 1 m^2^	Height, cm		Yield, t/ha			Drying Shrinkage	Dry Stems, t/ha
		Total	Without Flower Head	Total	Flower Head	Stems		
Marupe Municipality	134	96.32	84.20	16.59	10.32 (62%)	6.27 (38%)	31%	4.35
Adzelvieši, Burtnieki municipality	47	150.4	114.7	16.30	9.62 (59%)	6.67 (41%)	49%	3.43
Stariņi, Ogre municipality	57	138.6	107.1	18.68	11.29	7.39	54%	3.39

**Table 3 materials-16-05316-t003:** Produced hemp shive particleboard mean values of the physical properties.

Type	Thickness (mm)		Density (kg/m^3^)	
	Mean	Confidence Level	Mean	Confidence Level
HS-1	20.7	0.9	285	9
HS-2	21.2	1.1	283	12
MHF	24.5	0.8	274	7
CH	23.6	0.9	230	10
MHS	20.1	0.5	247	5

**Table 4 materials-16-05316-t004:** Material structure analysis with optical microscope.

Type	Surface Structure (a)	Sawn Cross Sections (b)	Sawn Edge (c)
HS-1	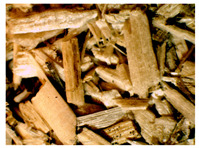	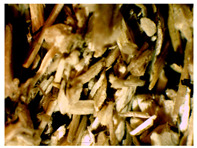	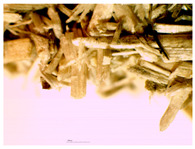
HS-2	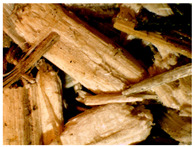	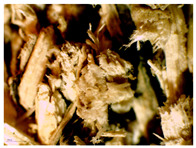	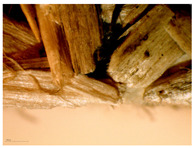
MHF	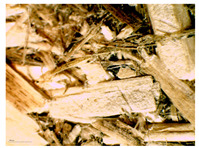	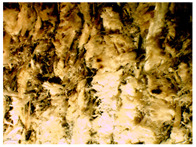	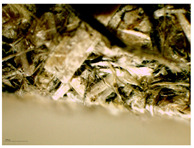
MHS	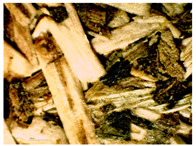	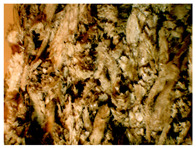	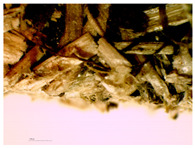
CH	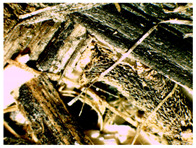	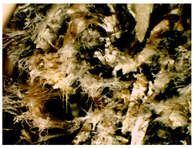	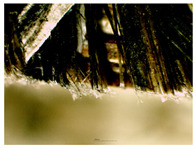

**Table 5 materials-16-05316-t005:** Material structure.

Name	Bulk Density, kg/m^3^	Porosity, %
Hemp shives (Raw material)	90	93.6
Hemp fibres (Raw material)	60	95.7
HS-1, HS-2	275	80.6
CH	200	86.1
MHF	375	72.8

**Table 6 materials-16-05316-t006:** Mechanical strength analysis of HS-1, HS-2, and CH board samples.

Type	Density, kg/m^3^		Bending Strength (MOR), MPa		MOE, MPa		IB, MPa		SW, N/mm	
	Mean	±	Mean	±	Mean	±	Mean	±	Mean	±
HS-1	275	4.62	2.4	0.34	197.2	31.71	0.08	0.017	10.6	1.6
HS-2	285	13.42	2.3	0.25	338.7	58.21	0.12	0.02	21.9	4.1
CH	230	15.97	3.5	0.55	231.4	90.56	0.08	0.005	13.5	2.6

## Data Availability

The data presented in this study are available in article. The measurement data sets for each presented evaluation in this study are available on request from the corresponding author.

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
