# Peer review of "Particleboard Creation from Agricultural Waste Residue of Seed Hemp"

_materials, 2023, doi:10.3390/ma16155316_

Round 1
Reviewer 1 Report
Good paper. Well organized and strctured. Following comments should be considered in the revised manuscript.
i) Compressive strength should be also measured.
ii) Correct the unit for density.
iii) Instead of heat conductivity coefficient use term thermal conductivity.
iv) The measurement uncertainty should be given.
v) Particle size distribution curves should be presented.
Author Response
Thank you for your time and effort to review the submitted article, please find the response to your points in review in attached document.

Reviewer 2 Report
The authors focus on an interesting and mainly current problematics, namely agricultural residues used for production of hemp-particleboard. In addition to the actual sowing of hemp, the conditions in the localities, the harvesting process, the preparation of the material and the production of different types of boards, their properties were subsequently examined. The article is primarily a follow-up to previous research by Zvirgzds et al. 2022 [11].
I have only a few minor remarks about the article and then rather personal considerations and observations.
In my opinion, the article in some passages is more like a "feasibility" study and it may not have been described in detail, e.g. the information on page 4.
Separating the results from the discussion has no effect, moreover, there is also a discussion in the chapter Results, so I propose to merge both chapters.
Check the numbering of chapters and sub-chapters, there are formal errors, e.g. 2.4.3, 3.3.4.
There are a significant number of abbreviations in the article, and they should always be explained when used within tables and graphs. Sometimes it wasn't even necessary to use them.
I have reservations about some of the technical terms used, e.g. "internal bond strength", but in the final consequence their meaning is all right.
For sure, I welcome considerations about feasibility, possible applications, and future research. Personally, I see more potential in “hemp-fiberboards”, but I understand that this process is not only generally more demanding, but also tied to much higher processing volumes, which would be a hindrance to local board production. However, the possibility of producing heat-insulating boards would be an advantage. What bothers me more is the environmental point of view, and that is the ingestion of UF resin, which I consider as not very appropriate. These boards will always have a higher absorbency compared to wood-based boards.
How do you explain the much higher MOE for HS-2 vs. HS-1 boards when the MOR is practically identical?
Author Response
Thank you for your time and effort to review the submitted paper.
Please find the information about the revisions done to manuscript in attached document and responses to your points and questions answered there.

Reviewer 3 Report
Very extensive (and partly too long) test report, not a scientific paper.
Despite all the details given, a real lot of questions occurred during the review, which must be clarified and answered (see attached paper).
Especially it is not clear which products are already on the market and what the improvements addressed in the paper really are.

Moderate improvement of style recommended.
Author Response
Thank you for your time and effort to review the submitted paper.
After receiving your and other two reviewer comments, the manuscript was revised to include your suggested input as much as possible.
Please find the responses to your comments into attached word document.

Round 2
Reviewer 3 Report
Paper has improved very much after the first review. Still some (new) small questions came up during the second review, see attachment.

Moderate editing recommended.
Author Response
Thank you again for your insight and very informative comments. The manuscript was revised accordingly. Our research group value your input and professional comments very highly.
Most of the rivisions are at places where you comments were at (Figure 4, lines 218 - 231; 506 - 508; 540 - 545; 613 - 618; 740 - 750). Please find the exact revisions in the manuscript file. The figures were revised with minor tweaks to the character size, font use and some decimal decreasin on the axis) Academic english editting was done up to chapter 2.2. The rest will be completed in coming days. We decided to upload now to speed up the revision process.
Please find the responses to tour comments in attached document (as replys to your original comments, I hope they are visible).
